# Modelling Malaria Incidence in the Limpopo Province, South Africa: Comparison of Classical and Bayesian Methods of Estimation

**DOI:** 10.3390/ijerph17145016

**Published:** 2020-07-13

**Authors:** Makwelantle Asnath Sehlabana, Daniel Maposa, Alexander Boateng

**Affiliations:** Department of Statistics and Operations Research, Private Bag X1106, University of Limpopo, Sovenga 0727, South Africa; asnath.sehlabana@ul.ac.za (M.A.S.); alexander.boateng@ucc.edu.gh (A.B.)

**Keywords:** malaria incidence, Bayesian estimation, classical estimation, Poisson regression model, negative binomial model

## Abstract

Malaria infects and kills millions of people in Africa, predominantly in hot regions where temperatures during the day and night are typically high. In South Africa, Limpopo Province is the hottest province in the country and therefore prone to malaria incidence. The districts of Vhembe, Mopani and Sekhukhune are the hottest districts in the province. Malaria cases in these districts are common and malaria is among the leading causes of illness and deaths in these districts. Factors contributing to malaria incidence in Limpopo Province have not been deeply investigated, aside from the general knowledge that the province is the hottest in South Africa. Bayesian and classical methods of estimation have been applied and compared on the effect of climatic factors on malaria incidence. Credible and confidence intervals from a negative binomial model estimated via Bayesian estimation and maximum likelihood estimation, respectively, were utilized in the comparison process. Overall assumptions underpinning each method were given. The Bayesian method appeared more robust than the classical method in analysing malaria incidence in Limpopo Province. The classical method identified rainfall and temperature during the night to be significant predictors of malaria incidence in Mopani, Vhembe and Waterberg districts. However, the Bayesian method found rainfall, normalised difference vegetation index, elevation, temperatures during the day and night to be the significant predictors of malaria incidence in Mopani, Sekhukhune and Vhembe districts of Limpopo Province. Both methods affirmed that Vhembe district is more susceptible to malaria incidence, followed by Mopani district. We recommend that the Department of Health and Malaria Control Programme of South Africa allocate more resources for malaria control, prevention and elimination to Vhembe and Mopani districts of Limpopo Province.

## 1. Introduction

Malaria is a mosquito borne disease caused by five protozoan species, namely: *Plasmodium Falciparum, Plasmodium vivax*, *Plasmodium malariae* and related species of *Plasmodium ovale* and *Plasmodium knowlesi* ([1]). The protozoa are transmitted to humans through the bites of infected female *Anopheles* mosquitos (mosquitos carrying protozoa). *Plasmodium falciparum* is known to account for many malaria cases globally and is therefore regarded as a threat to public health worldwide ([1,2]). Malaria incidence refers to the commonness of malaria occurrence. When the incidence rates are high, transmission and prevalence of malaria are also high. This exposes the vulnerability and danger of the disease to society.

The symptoms of malaria include fever (>37.5 °C), headache, rigors, muscle pains, diarrhea, nausea, vomiting, loss of appetite, inability to feed babies, dizziness and sore throat. Based on history, malaria has infected and taken the lives of millions of individuals. This disease remains a major cause of human morbidity and mortality in most of the developing countries in Africa. Young children, pregnant women, and elderly individuals are groups of people at higher risk of malaria transmission ([3]). Sachs and Malaney [4] outlined the factors that contribute to increased malaria cases. These encompassed changing agricultural practices, building of more dams, poor irrigation skills, deforestation, poor public health services and long-term climate change causes such as El Nino and global warming. Hay et al. [5] found seasonal climatic change to be an important determinant of malaria incidence since variations in climate conditions could increase mosquito vector dynamics and parasite development rates ([6,7]). Indeed, malaria incidence has been found to be generally low during dry-hot season when vector populations are reduced and spatially restricted. As a result, several of studies on malaria incidence tend to focus on the peak transmission season, which is often the rainy season, whereas the epidemiological picture during the dry-hot season is often neglected ([7,8]).

According to Blumberg and Frean [9], there has been great progress in malaria control globally. This progress is attributed to increased funding, improved use of life-saving interventions and more countries pursuing malaria elimination measures. Although the progress achieved in countries such as Sri Lanka and some Sub-Saharan African countries has been considerable, South Africa remains among countries with high risk of malaria transmission, especially the northern part of the country ([7,8,10]). Raman et al. [10] further outlined that South Africa officially transitioned from controlling malaria to the goal of eliminating the disease in 2012. However, malaria cases have increased from 6811 in 2013 to 11,711 in 2014, with many cases reported in the Mpumalanga and Limpopo provinces of South Africa ([7]). It will therefore be expedient to model malaria incidence in Limpopo Province because it is amongst the provinces that account for most malaria cases in South Africa ([7,8]). The present study will employ both classical and Bayesian methods of estimation to assess the effect of climatic factors such as temperature, rainfall, elevation and normalised difference vegetation index on malaria incidence. The results of the present study may well assist in malaria control programs for inspection, control, prevention and possible elimination of malaria in Limpopo Province.

This study is crucial because there are still arguments concerning association between climatic factors and malaria incidences ([11]). Yé et al. [11] highlighted that effects of climatic factors on malaria transmission are not efficiently assessed, specifically at local levels. Yé et al. [11] further outlines that data used in many studies are proxy meteorological data obtained through satellites or interpolated from a different scale. To the best of our knowledge limited or no prior study on malaria prevalence in the province or the country has employed local scale data. Hence, in this present study, a local scale data from a malaria control institution in Limpopo province will be used. Indeed, the gap that we seek to address in this paper does not only lie in the data used in fitting the model but also in the methodology. For instance, environmental factors vary overtime hence classical methods may not do “justice” to the data, hence the introduction of the Bayesian methods. This is absent in many of the previous studies cited earlier.

In several studies that modelled malaria counts, methods such as Poisson, negative binomial, hurdle, quasi-Poisson and dynamic computable general equilibrium (DCGE) models have been applied. For instance, Shimaponda-Mataa et al. [12,13,14,15] modelled the environmental factors and assessed their relationships with malaria incidence through the development of Poisson regression models. Kazembe [16] conducted a research on malaria incidence and found negative binomial regression model to provide a better fit compared to Poisson regression model. In other separate studies, Shimaponda-Mataa et al. [12,17,18] found a positive relationship between rainfall and malaria risk. On the contrary, Zayeri et al. [14] found a negative relationship between rainfall and malaria incidence. Studies by Shimaponda-Mataa et al. [12,18] further revealed a positive relationship between minimum temperature and malaria risk. Gerritsen et al. [14,19] provided evidence that adults are more susceptible to malaria transmission than children. There are also studies that modelled malaria incidence using other different methods. These methods include the time series data analysis methods employed in the studies of Adeola et al. [20,21], and the qualitative retrospective descriptive method employed in the study of Machimana [22]. The studies by Adeola et al. [20,21] found that malaria cases have a positive relationship with both temperature and rainfall. These results support the findings of Machimana [22], which revealed that malaria transmission in South Africa is associated with climate. Machimana [22] study further revealed that malaria cases are highly seasonal, with higher number of cases in January to April and October. The findings of the study by Machimana [22] also indicated that persons between the ages 16 to 40 years and males are more susceptible to malaria transmission.

Abiodun et al. [8] conducted a study on the resurgence of malaria prevalence in South Africa between 2015 and 2018. Their study concentrated on reviewing several malaria-related research articles that were published using Arksey and O’Malley framework. Out of a total of 534 malaria related articles that were reviewed, very few of them made use of Bayesian estimations. The argument in the present study is that climatic variables in relation to malaria prevalence are dynamic and classical statistics estimators such as maximum likelihood methods will probably not be efficient compared to Bayesian estimation, where climatic variables are treated as random variables with some underlying distribution. In another study, Abiodun et al. [7] conducted a study on malaria incidence in Limpopo Province using dynamical and zero-inflated negative binomial regression models. Results from their study revealed the effect of rainfall and average temperature on malaria incidence. The present paper is different from several previous papers especially in terms modelling and parameter estimation.

Another point of departure is that many of the data sets employed in most of these previous studies in one way or the other have used environmental factors and other predictors of malaria. Environmental factors such as elevation, temperature, normalised difference vegetation index (NDVI) are dynamic and their effect on malaria incidence cannot be fixed, although they are unknown as claimed by the classical approach. A Bayesian estimation therefore will be adopted in the present study to model these factors on malaria incidence because it considers these environmental factors as random variables with some probability distribution. This has made the use of Bayesian estimation methods very popular. In classical paradigm, parameters of a model are unknown, but fixed constants, while in Bayesian estimation the parameters are random, with knowledge about the parameters described in the form of a probability distribution. We seek to explore these two methods and compare them in the context of malaria incidence in Limpopo Province of South Africa. This is limited in a number of studies conducted on malaria incidence, especially in Southern Africa.

In this paper, classical and Bayesian estimation methods will be utilised and compared in the context of modelling malaria incidence. The relation between the two statistical estimations are from the fact that the posterior distribution in the Bayesian approach is proportional to the likelihood function times the prior distribution. Whereas maximum likelihood estimation (MLE) uses asymptotic distributional assumptions in classical statistics, the uncertainty about model parameters in the Bayesian approach is expressed through the prior distributions. Combining the prior distribution and likelihood (data), researchers are able to update the knowledge about the model parameters. This is done through the posterior distribution from which we can infer the estimates of the model parameters and relevant quantities like credible intervals.

The rest of the paper is outlined as follows: Section 2 describes materials and methods while Section 3 presents the discourse on the results. Discussion and concluding remarks are presented in Section 4 and Section 5, respectively.

## 2. Materials and Methods

### 2.1. Study Frame and Data Collection

This study models malaria incidence in Limpopo Province of South Africa. Limpopo Province consists of five districts: Capricorn, Mopani, Sekhukhune, Vhembe, and Waterberg. Malaria incidence or cases data were provided by Malaria Control Center located in Tzaneen. The population data were provided by Statistics South Africa (StatsSA) [23]. Environmental factors: rainfall, temperature, elevation, and normalised difference vegetation index data were obtained from Ecoverb. The data were collected monthly from January 2014 to June 2015. R software was used in analysing the data while the bar charts were done in Excel.

#### 2.1.1. Classical Methods

##### Poisson Regression Model

The Poisson distribution is probably the most used discrete distribution because of its simplicity. The Poisson probability mass function is given by:(1)$$E\left(Y=y\right)=\frac{\lambda^{y}e^{-\lambda }}{y!},y=0,1,2,\ldots$$

The mean and variance of the Poisson distribution are equal to λ. Hence, for Poisson regression, we have:(2)$$E(Y_{i})=\lambda =\mu (x_{i})=e^{\beta_{0}+\beta_{1}x_{i1}+\beta_{2}x_{i2}+\ldots +\beta_{k-1}x_{i,k-1}}.$$

Consequently, the Poisson regression model is:(3)$$P(X=y_{i}|x_{i})=\frac{{\left[\mu (x_{i})\right]}^{y}e^{-\mu (x_{i})}}{y!},y=0,1,2,\ldots$$

From (3), the mean and variance of the Poisson regression model are equal to $\mu (x_{i})$. Thus, the Poisson regression is apt for count data where the mean and variance are numerically identical. The values of a count response variable y are non-negative. Therefore, the mean function $\mu (x_{i})$ safeguards the non-negative nature of the response variable y.

##### Negative Binomial (NB) Model

The use of negative binomial (NB) model in count data modelling, as it is the case in this study, often comes up when there is over-dispersion. While the Poisson distribution is often first to be considered for fitting count data such as malaria incidence, nevertheless, if the mean is very much less than the variance of the data, then there is over-dispersion in the data. An alternative approach to correct over-dispersion is to fit a NB regression model to the over-dispersed Poisson regression model. The probability mass function for the NB distribution is given by:(4)$$P(Y=y)=\left(\begin{array}{c} \frac{1}{\alpha }+y-1\\ y \end{array}\right)\lambda^{y}{\left(1-\lambda \right)}^{\frac{1}{\alpha }},y=0,1,2\ldots$$

The mean and variance of the above model are respectively given by $\mu =\frac{\lambda }{\left[\alpha (1-\lambda )\right]}$ and $\sigma^{2}={\left[\frac{\lambda }{\alpha [1-\lambda ]}\right]}^{2}$. From the mean and variance, we can envisage that the NB distribution is over-dispersed since the variance surpasses the mean. Assume the mean of the NB distribution depends on some predictor variable $x_{i}$, then we can write the mean $\mu (x_{i})=\frac{\lambda }{\left[\alpha (1-\lambda )\right]}$ from which λ is obtained as $\lambda =\frac{\alpha \mu (x_{i})}{\left[1+\alpha \mu (x_{i})\right]}$. Hence, the NB regression model can be expressed as:(5)$$P(Y=y_{i}|x_{i})=\left(\begin{array}{c} \frac{1}{\alpha }+y_{i}-1\\ y_{i} \end{array}\right){\left(\frac{\alpha \mu (x_{i})}{1+\alpha \mu (x_{i})}\right)}^{y_{i}}{\left(\frac{1}{1+\alpha \mu (x_{i})}\right)}^{\frac{1}{\alpha }},y=0,1,2,\ldots$$

The mean and variance of the NB regression model are correspondingly given by $E(Y)=\mu (x_{i})$ and $V(Y)=\mu (x_{i})[1+\alpha \mu (x_{i})]$. The NB regression model reduces to the Poisson regression model when the dispersion parameter α=r goes to zero ([24]).

##### Maximum Likelihood Estimation

Assume we observe $y_{i}(i=1,2,\ldots ,n),$ count response variables, each with predictor variables $x_{i1},x_{i2},\ldots ,x_{i,k-1}$. The likelihood function for the Poisson regression model is obtained by multiplying the respective probabilities in (3) to obtain:(6)$$L(\beta_{0},\beta_{1},\ldots ,\beta_{k-1})=\prod_{i=1}^{n}\frac{{\left[\mu (x_{i})\right]}^{y_{i}}e^{-\mu (x_{i})}}{y_{i}!}$$

Taking the log of (6) gives the log-likelihood function as:(7)$$\log(L)=L(\beta_{0},\beta_{1},\ldots ,\beta_{k-1})=\sum_{i=1}^{n}\left\{y_{i}\log[\mu (x_{i})]-\mu (x_{i})-\log(y_{i}!)\right\}$$

Similarly, the log-likelihood function for NB regression model can also be estimated via maximum likelihood. Cameron and Trivedi [24] gives the logarithm function as:(8)$$\log(L)=\sum_{i=1}^{n}\left\{\begin{array}{l} \log[\Gamma (y_{i}+\alpha^{-1})]-\log[\Gamma (\alpha^{-1})]-\log[\Gamma (y_{i}+1)]\\ -\alpha^{-1}\log(1+\alpha \mu_{i})-\\ y_{i}\log(1+\alpha \mu_{i})+y_{i}\log(\alpha )\\ +y_{i}\log(\mu_{i}) \end{array}\right\}$$

A common goodness-of-fit statistic for count regression models is the Pearson’s $\chi^{2}$ statistic defined by:(9)$$\chi^{2}=\frac{\sum_{i=1}^{n}{\left(y_{i}-\overset{\wedge }{\mu_{i}}\right)}^{2}}{v(\overset{\wedge }{\mu_{i}})}$$
where $\overset{\wedge }{v(\mu_{i})}$ is the variance function assessed at the estimated mean. The log-likelihood in (7) and (8) can be used as goodness-of-fit statistic. However, the log-likelihood and the Pearson’s chi-square displayed in Equation (9) do not take into account the number of estimated parameters in the model, hence the use information criteria such as Akaike Information Criterion (AIC) become necessary. From the NB regression model, the number of estimated parameters is given by p*=p+1. The extra 1 is from the dispersion parameter in NB regression model. In this study, we use the AIC as a goodness-of -fit statistic, which take into consideration p*, the number of parameters. Generally, the higher the number of parameters, the greater the log-likelihood, while AIC penalizes for the number of parameters and it is given by:(10)AIC=−2log(L)+2p*

#### 2.1.2. Bayesian Approach

Bayesian statistics to a large extent can be attributed to Reverend Thomas Bayes (1701–1761), who developed Bayes’ theorem. Bayes’ theorem expresses the conditional probability, or *posterior* probability, of an event A after B as observed in terms of the *prior* probability of A, *prior* probability of B, and the conditional probability of B given A. The foundation for Bayesian inference is defined from Bayes theorem and is given as follows:(11)$$P_{r}\left(A|B\right)=\frac{P_{r}(B|A)P_{r\;}\left(A\right)}{P_{r\;}\left(B\right)}$$

By substituting B with observations y, A with parameter set or space Θ and probabilities $P_{r\;}$ with densities p, Equation (11) becomes:(12)$$p\left(\Theta |y\right)=\frac{p(y|\Theta )p_{\;}\left(\Theta \right)}{p\left(y\right)}$$
$$\left(i.e.\right)\;\;\;\;p\left(\Theta |data\right)=\frac{p(data|\Theta )p\left(\Theta \right)}{\int p(data|\Theta )p\left(\Theta \right)d\Theta_{\;}}$$
where p(y) is the marginal likelihood of *y*, p(Θ) is the set of prior distributions of parameter set Θ before y is observed, p(y|Θ) is the likelihood of y under a model, and p(Θ|y) is the joint posterior distribution of the parameter set or space Θ that expresses the uncertainty about j parameter set Θ after taking the prior and the data into account. Recall that $\Theta =\theta_{1},\ldots ,\;\theta_{j}$ with denominator expressed as:(13)p(y)=∫p(y|Θ)p(Θ)dΘ,
where defines the marginal likelihood of  y or the prior predictive of y, and may be set into an unidentified constant c resulting in the following:(14)$$p\left(\Theta |y\right)=\frac{p(y|\Theta )p\left(\Theta \right)}{c}.$$

The presence of the marginal distribution likelihood of y normalises the joint posterior distribution  p(Θ|y), ensuring it is a proper distribution and integrates to one. Eliminating the constant  c from Equation (13) will result in some changes in the relationship from the use of the equal sign to the constant of proportionality, resulting in the following:(15)p(Θ|y)∝p(y|Θ)p(Θ)

#### 2.1.3. Computation of NB Using Bayesian Estimation

Through the Markov chain Monte Carlo (MCMC), we use a Gibbs sampler for the NB regression model. This model is derived from the Poisson regression model to account for over-dispersion, which usually happens or occurs in count data. Suppose the response are independent, then:$$Y_{i}\sim Negbin(\lambda_{i},r)$$
where $Y_{i}$ is the response variable for i=1,2,…,n; r is the over-dispersion parameter. The expectation is modelled as:(16)$$\log(\lambda_{i})=X_{i}^{T}\vec{\beta },$$
which implies that:(17)$$\lambda_{i}=\exp(X_{i}^{T}\vec{\beta })$$
where X is the matrix of regressors and $\vec{\beta }$ is the parameter vector.

The conditional likelihood of $Y_{i}$ given $w_{i}$ is defined as:(18)$$L(Y_{i}|r,\vec{\beta },w_{i})\propto \exp\left\{k_{i}X_{i}^{T},\vec{\beta }-{\left(X_{i}^{T}\vec{\beta }\right)}^{2}/2\right\}\propto \exp\left\{\frac{-w_{i}}{2}{\left(\frac{y_{i-}-r}{2w_{i}}-X_{i}^{T}\vec{\beta }\right)}^{2}\right\}$$
where $k_{i}=\frac{y_{i}-r}{2}$. Now, exploiting property 1 of the poly-Gamma (PG) distribution, Equation (18) can be written as:(19)$$L(Y_{i}|r,\vec{\beta },w_{i})=e^{k_{i}\eta_{i}}\int_{0}^{\infty }e^{-\psi_{i}\frac{\eta_{i}^{2}}{2}}p(\psi_{i}|r,Y_{i}0)d\psi_{i}$$
where $\eta_{i}=X_{i}^{T}\vec{\beta }$. Suppose $\psi_{i}$ is distributed according to poly-Gamma (PG) ~ $\left(Y_{i}+r,\eta_{i}\right)$, then following Scott and Pillow (2012), the conditional probability for $\vec{\beta }$ is given by:(20)$$p(\vec{\beta }|\vec{Y}*,r,\vec{w}\vec{\psi })\propto \pi (\vec{\beta })\exp\left[\frac{-1}{2}{\left(z_{i}-X^{*}\vec{\beta }\right)}^{T}\left(z-X^{*}\vec{\beta }\right)\Omega \right]$$
where $\vec{Y}*$ is equal to the n*1 subvector of $\vec{Y}$ corresponding to $w_{i}$; $n*=\sum_{i=1}^{n}w_{i}$ is the number of individuals in risk class; $\vec{\psi }$ is a vector of length n* with elements $z_{i}=\frac{y_{i}-r}{2\psi_{i}}$; $\Omega diag(\psi_{1},\ldots ,\psi_{n})=n*n$ is the precision matrix and X*=N*×P matrix. From Equation (20), it is clear that $\vec{z}$ is normally distributed with mean $\vec{\eta }=X*\vec{\beta }$ and diagonal covariance matrix $\Omega^{-1}$. Therefore, it is reasonable to assume a conditional Gaussian prior for $\vec{\beta }$ denoted by:$$N_{p}\left(\vec{\beta_{0}},\sum_{0}\right)$$

The conjugate prior full conditional distribution for $\vec{\beta }$ given $\vec{z}$ and Ω follows $N_{p}\left(\vec{\mu },\sum \right)$, where $\sum ={\left(\sum_{0}^{-1}+X{*}^{T}\Omega X*\right)}^{-1}$ and $\vec{\mu }=\sum \left(\sum_{0}^{-1}+X{*}^{T}\Omega \vec{z}\right)$. Therefore given the current values for $\vec{\beta },\vec{w}$ and r, the Gibbs sampler is given as follows:
for $w_{i},$ draw $\psi_{i}$ from its PG ~$\left(Y_{i}+r,\eta_{i}\right)$ distributionfor $w_{i},$ define $z_{i}=\frac{y_{i}-r}{2\psi_{i}}$,update $\vec{\beta }$ from $N_{}\left(\vec{\mu },\sum \right)$ distribution,update r using a random-walk Metropolis-Hastings algorithm.

## 3. Results

This section presents and discusses the results for fitting count regression models, namely; the Poisson regression model and NB model estimated with maximum likelihood, and Bayesian estimation methods. The variables used in fitting both models are described in Table 1, with descriptive statistics presented in Table 2. The distribution of malaria incidence in Limpopo province is presented in Figure 1. The posterior results for the Bayesian method are presented in Appendix A.

The histogram displayed in Figure 1 shows that the distribution of malaria incidence is skewed to the right. This implies that it takes a lopsided mound shape with its tail going off to the right. The shape of this histogram is similar to the shape of a Poisson distribution. Hence we model the data using Poisson regression model.

According to Figure 2, the transmission rate of malaria was high in 2014 than in 2015. This probably may be due to various effects of environmental factors as they may differ in each year and it may also indicate the success of malaria control, prevention and elimination methods that are used currently in Limpopo Province.

As shown in Figure 3, Vhembe district is depicted to have the highest rate of malaria incidence, followed by Mopani district as compared to all the other districts. Capricorn district has the lowest rate of malaria incidence. The high rate of malaria incidence in Mopani and Vhembe districts could be attributed to the high temperatures in the two districts.

The *p*-values for all the covariates as displayed in Table 3 are less than the level of significance, 0.05. This suggests evidence against the null hypothesis of no relationship between the covariates and malaria incidence. Rainfall and elevation estimate values are negative, suggesting that they have a negative relationship with malaria incidence. Table 3 also depicts a positive relationship between temperature, NDVI and malaria incidence. When the ratio of the deviance statistic and its degrees of freedom is significantly larger than 1, then there is evidence of lack of fit in the model developed. The ratio of the deviance statistic and its degrees of freedom is 12.753, which is significantly larger than 1. Hence there is evidence of lack of fit for the model presented in Table 3.

### Detection of Over-Dispersion

The Pearson’s chi-square is considered to be robust in detecting over-dispersion in Poisson models. If the ratio of the residual deviance and the degree of freedom is significantly larger than 1, then the probability that the developed model is over-dispersed is high. Based on the Poisson model, the ratio of the residual deviance and the degrees of freedom is 12.753. This implies that the probability that the selected Poisson model is over-dispersed is very high. To validate that the Poisson model selected may be over-dispersed, we check if the response variable satisfies the Poisson assumption of an equality between the mean and the variance. Table 2 shows that the mean for the response variable is 23.95 and the variance is found to be 3822.5. Therefore, the condition of equal mean and variance for a Poisson distribution is violated. We can then conclude that the Poisson model presented by Table 3 is over-dispersed.

Table 4 presents the NB model developed to correct the overdispersed Poisson model presented by Table 3. According to Table 4, the 95% confidence interval for the covariate rain does not include 0, which implies that it is significant at 5% level of significance. Hence, there is a relationship between rainfall and malaria incidence. The coefficient estimate of rain is negative. This implies that the relationship between rainfall and malaria incidence is negative. That is, malaria transmission rate increases with a decreasing amount of rainfall. The *p*-values for Mopani, Vhembe and Waterberg are very close to 0. These *p*-values implies that there is a certain pattern of malaria transmission between these districts and Capricorn district (the reference category). The coefficient estimates for Mopani, Vhembe and Waterberg are positive. These estimates entail that if malaria incidence increases in each of these districts, then it also increases in Capricorn district (the reference category). We are using the odds ratio, $e^{\beta }$ to find the precise pattern of malaria incidence amongst the districts. The odds ratio in this case is the ratio of the odds of the reference category (Capricorn) and each of the districts Mopani, Vhembe and Waterberg. If there is an increase in malaria incidence, the increase is $e^{\beta }$ times more in Mopani, Vhembe and Waterberg than in Capricorn district. The Greek letter *β* in the odds ratio $e^{\beta }$ represents the regression coefficient. Table 4 provides evidence that malaria incidence increases by *e*^2.215^ ≈ 9 times in Mopani, *e*^2.848^ ≈ 17 times in Vhembe and *e*^0.8711^ ≈ 2 times in Waterberg than in Capricorn district. A unit increase in temperature during the night increases the incidence of malaria by *e*^0.2537^ ≈ 1 unit. There is no evidence of an existing association between malaria incidence and the covariates, temperature during the day, elevation and NDVI according to Table 4. The ratio of the deviance statistic and its degrees of freedom is 1.148, which is significantly close to 1 compared to the ratio of the deviance statistic and its degrees of freedom for the Poisson model presented in Table 3. Hence there is an evidence of good fit in the model presented in Table 4.

All the 95% credible intervals presented in Table 5 do not include zero, which indicates that all the variables are significant, except for Waterberg. However, the variable of NDVI is extremely significant while other parameters are moderately significant. This implies that malaria incidence is affected more by NDVI than other environmental factors. Both 95% highest posterior density (HPD) credible intervals for the regression coefficients of the covariates rainfall and elevation are negative. 

This implies that there is a very high probability that the estimates of these regression coefficients are negative. Therefore, we can conclude that the relationship between malaria incidence and each of the covariates rain and elevation is negative. That is, an increase in rainfall leads to a decrease in malaria incidence and an increase in elevation above sea level leads to a decrease in malaria incidence.

All the 95% HPD credible intervals for temperature during the night, temperature during the day and NDVI are positive, which indicate that there is a very high probability that the estimates of these regression coefficients are positive. Therefore, we can conclude that the relationship between each of these covariates and malaria incidence is positive. That is, an increase in temperature during the night, temperature during the day and NDVI results in an increase in malaria incidence.

The 95% HPD credible intervals for Mopani, Sekhukhune and Vhembe districts are positive, which indicate that as malaria incidence increase in each of these districts, it also increases in Capricorn district (reference variable). However, both the 95% HPD credible intervals for Waterberg are negative. This implies that if malaria incidence increases in Capricorn district, then it decreases in Waterberg district. We can then conclude, according to the MCMC estimation methods applied to obtain the model in Table 5 that there is a relationship between malaria incidence and each of the environmental factors included in this study.

## 4. Discussion

Both the Bayesian and classical methods revealed a positive relationship between malaria incidence and temperature during the night. That is, an increase in temperature during the night results in an increase in malaria incidence. Therefore, we can conclude that the risk of malaria transmission is high during warm nights, which are usually the nights of summer seasons. The Bayesian and classical frameworks produced similar results about the relationship between malaria incidence and rainfall, which was found to be negative. Therefore, we can conclude that an increase in the amount of rainfall results in a decrease in malaria incidence. The classical framework does not provide any evidence of an existing relationship between malaria incidence and either elevation, temperature during the day or NDVI. However, the Bayesian framework revealed that an increase in NDVI and temperature during the day lead to increased malaria incidence while an increase in elevation above sea level leads to decreased malaria incidence. Malaria incidence increases in Mopani and/or Vhembe districts, then it also increases in Capricorn district. The classical framework revealed no pattern of malaria incidence between Capricorn and Sekhukhune districts while the Bayesian framework suggests that if malaria incidence increases in Sekhukhune district, then it also increases in Capricorn district. The Bayesian framework also suggests that if malaria incidence decreases in Waterberg (though not significant) district then it increases in Capricorn district while in contrast, the classical framework suggest that if malaria incidence increases in Waterberg district then it also increases in Capricorn district. Both methods affirm that Vhembe district is more susceptible to malaria incidence, followed by Mopani district, confirming the results of Abiodun et al. [7,8]. The classical method did not identify any particular trend of malaria incidence over the period of study. However, the Bayesian method identified an upward trend of malaria incidence over the period of the study. Again, the MLE method generated more errors and wider intervals while the Bayesian estimation method generated fewer errors and narrower intervals. Therefore, we can conclude that the Bayesian method of estimation outperforms the classical method of estimation.

## 5. Conclusions

This study was limited to one South African province, Limpopo. However, there is another province, Mpumalanga which is also known to account for high malaria cases in South Africa. This study only made use of non-informative priors. However, future research may consider Bayesian estimation under dissimilar prior distributions such as improper, conjugate and Jeffrey’s priors in the model fitting process. The spatial dimension in the context of Bayesian estimation may be considered in analysing the prevalence of diseases such as malaria in future research.

Based on the findings from this study, we recommend that the Department of Health and Malaria Control Programme of South Africa allocate more resources for malaria prevention, control and elimination to Vhembe and Mopani districts in Limpopo Province. We also recommend that the government provide educational seminars to educate the South African communities on how to prevent malaria transmission, especially during the warm summer nights.

## Figures and Tables

**Figure 1 ijerph-17-05016-f001:**
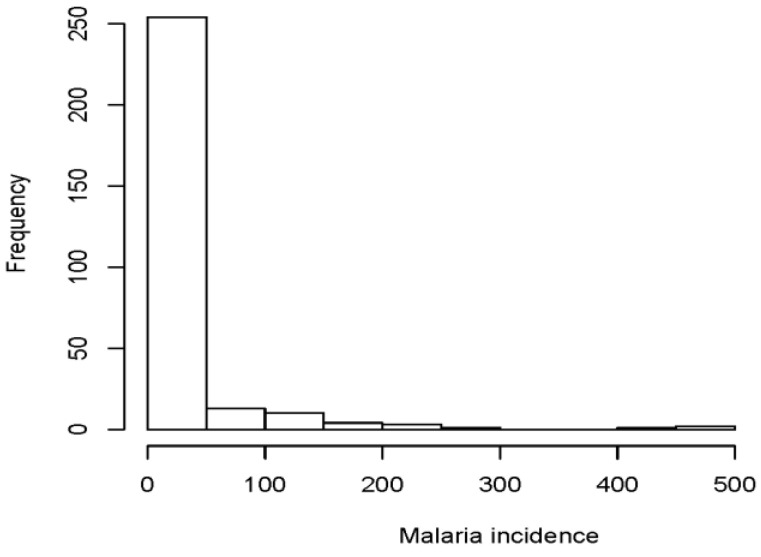
The distribution of malaria incidence in the Limpopo province.

**Figure 2 ijerph-17-05016-f002:**
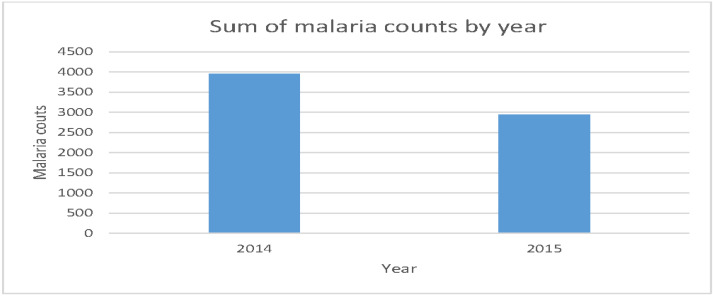
The distribution of malaria incidence in 2014 and 2015.

**Figure 3 ijerph-17-05016-f003:**
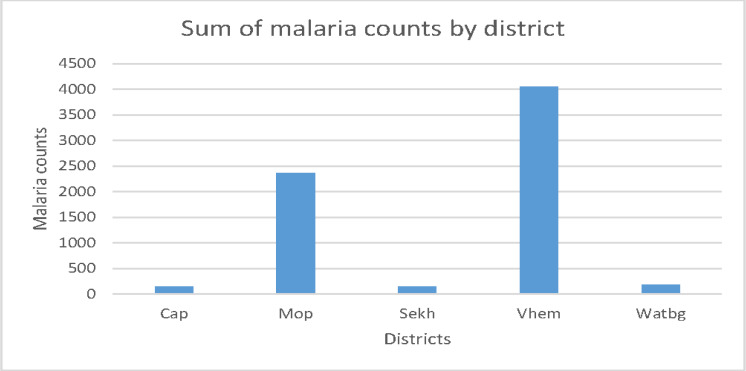
The distribution of malaria incidence across the districts of Limpopo province.

**Table 1 ijerph-17-05016-t001:** Variable description.

Variable	Description	Data Set Code
Malaria	The number of malaria cases.	*mal*
Population	The population size.	*pop*
Districts	The five districts of Limpopo province: Capricorn, Mopani, Sekhukhune, Vhembe and Waterberg.	*dist*
Years	The data for which the data were collected: 2014 and 2015.	*dyear*
Elevation	The elevation above sea level measured in meters.	*ele*
Rainfall	The rainfall measured in millimetres.	*rain*
NDVI	The difference between near-infrared reflected by the vegetation and red light which is absorbed by the vegetation, it ranges from −1 to 1.	*ndvi*
Temperature during the day	The maximum temperature in degrees Celsius.	*td*
Temperature during the night	Minimum temperature in degrees Celsius.	*tn*

**Table 2 ijerph-17-05016-t002:** Descriptive Statistics.

Statistics	Ele	Tn	Td	NDVI	Rain	Mal
Min	19.690	4.920	20.990	0.210	0.000	0.000
Max	822.241	25.600	40.240	0.660	159.200	4820.00
Mean	325.270	16.250	31.660	0.410	32.591	23.95
Median	242.200	16.710	32.441	0.400	23.910	3.000
Std. Error	216.090	4.450	4.151	0.110	38.420	61.830
1st Quartile	182.431	13.170	28.720	0.320	0.520	1.000
3rd Quartile	491.181	19.550	34.640	0.490	47.021	12.250

**Table 3 ijerph-17-05016-t003:** Parameter estimates for Poisson regression model under classical approach encompassing all the variables.

Coefficient	Estimate	Std. Error	95% Confidence Intervals
Intercept	−17.850	0.288	[−18.414, −17.288] **
Rainfall	−0.008	0.000	[−0.009, −0.007] **
Province (Ref: Capricorn)	-	-	-
Mopane	2.213	0.089	[2.048, 2.385] **
Sekhukhune	0.537	0.122	[0.297, 0.777] **
Vhembe	2.458	0.084	[2.297, 2.627] **
Waterberg	0.489	0.111	[0.272, 0.708] **
Year (Ref: 2014)	-	-	-
2015	−0.170	0.029	[−0.227, −0.114] **
Temperature during the night	0.276	0.007	[0.263, 0.290] **
Temperature during the day	0.064	0.008	[0.049, 0.080] **
Elevation	−0.001	0.000	[−0.001, −0.001] **
Normalized Difference Vegetation Index	0.521	0.228	[0.074, 0.969] **
Deviance			DF
Null Deviance	18541.3	-	287
Residual Deviance	3532.6	-	277
Dispersion parameter (α/r)	12.753	-	-
AIC	4421.4	-	-

NB: ** Indicates the significance of the variable within [95%] confidence interval for the five districts in Limpopo province where the study was conducted.

**Table 4 ijerph-17-05016-t004:** Parameter estimates for negative binomial (NB) model under classical approach encompassing all the variables.

Coefficient	Estimate	Std. Error	95% Confidence Intervals
Intercept	−1.3300	1.0380	[−17.4827, –13.1860] **
Rainfall	−0.0051	0.0020	[−0.0095, −0.0006] **
Province (Ref: Capricorn)	-	-	-
Mopani	2.2150	0.2086	[1.7890, 2.6425] **
Sekhukhune	0.4149	0.2577	[−0.0966, 0.9249]
Vhembe	2.8480	0.2113	[2.4123, 3.2851] **
Waterberg	0.8711	0.2291	[0.3950, 1.3484] **
Year (Ref: 2014)	-	-	-
2015	0.2056	0.1251	[−0.0398, 0.4521]
Temperature during the night	0.2537	0.0325	[0.1814, 0.3264] **
Temperature during the day	0.0000	0.0328	[−0.0705, 0.0704]
Elevation	−0.0005	0.0005	[−0.0014, 0.0005]
Normalized Vegetation Index	0.4769	1.0140	[−2.5551, 1.6125]
Deviance			DF
Null Deviance	1232.31	-	287
Residual Deviance	317.87	-	277
Dispersion parameter (α/r)	1.148	-	-
AIC	1680.9	-	-

NB: ** Indicates the significance of the variable within [95%] confidence interval for the five districts in Limpopo Province where the study was conducted.

**Table 5 ijerph-17-05016-t005:** Parameter estimates for negative binomial (NB) model with Bayesian approach encompassing all the variables.

Coefficient	Estimate	Naive Std. Error	95% HPD Credible Intervals
Intercept	−8.179	0.005	[−8.830, −7.516] **
Rainfall	−0.001	0.000	[−0.003, −0.000] **
Province (Ref: Capricorn)	-	-	-
Mopani	2.169	0.001	[1.994, 2.329] **
Sekhukhune	0.335	0.002	[0.090, 0.562] **
Vhembe	2.817	0.001	[2.664, 2.992] **
Waterberg	−0.219	0.002	[−0.454, −0.006]
Year (Ref: 2014)	-	-	-
2015	0.086	0.001	[0.013, 0.157] **
Temperature during the night	0.111	0.000	[0.093, 0.129] **
Temperature during the day	0.165	0.000	[0.146, 0.184] **
Elevation	−0.004	0.000	[−0.001, −0.000] **
Normalized Vegetation Index	4.911	0.004	[4.343, 5.457] **

NB: ** Indicates the significance of the variable within [95%] HPD credible interval for the five districts in Limpopo province where the study was conducted.

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
