# Peer review of "Modelling Malaria Incidence in the Limpopo Province, South Africa: Comparison of Classical and Bayesian Methods of Estimation"

_ijerph, 2020, doi:10.3390/ijerph17145016_

Round 1
Reviewer 1 Report
Although an interesting study, the authors do not show addition of any new information. Several studies already show that in South Africa, malaria is mainly found in three epidemic provinces, Limpopo, Mpumalanga and KwaZulu-Natal e. g. Gbenga J. Abiodun1 et al. 2020. Investigating the Resurgence of Malaria Prevalence in South Africa between 2015 and 2018: A Scoping Review; National Department of Health (NDoH) - Annual Report. 2018. Other studies have considered the connections between malaria incidence and climate variables using mathematical and statistical models, e.g. Gbenga et al. 2019. A dynamical and zero-inflated negative binomial regression modelling of malaria incidence in Limpopo province, South Africa. Interestingly none of these studies were cited in this study.
The authors should carefully go through the above cited manuscripts and show how this adds to what is already known. In addition, they should use malaria models that integrate all variables to monitor the progression of malaria and assist in intervention and prevention efforts.
Additionally, the models could be implemented for developing methodologies necessary to detect malaria pathogen, vector and habitat preference, or as a framework for strengthening the institutional capacity of malaria surveillance by providing malaria-risk management based on climate information and early warning system.
In addition, the authors should pay close attention to language structure and follow reference guidelines of the Int. J. Environ. Res. Public Health. Some examples in text include:
- Lines 38-39 and throughout - Scientific names are always italicized and specific names not capitalized e.g. Plasmodium falciparum, Plasmodium vivax, Plasmodium malariae,
- Lines 51, 53, 65, etc.. – use of references in text must follow journal format. You cannot begin a sentence by …..[4] outlined factors that contribute to increased malaria cases…..
- Line 60 for example should read, “According to Blumberg et al. (2017), there is a fairly great progress in malaria control globally.” There are numerous examples like these that need to be corrected in text
Reviewer 2 Report
Overall
The authors conducted a study to compare two statistical procedures in the modeling of malaria incidence in a South African province. The study could be of interest to investigators and most of the general population in South Africa. Overall, the paper needs to be shortened. It also needs to have English language editing as several sentences are awkwardly written. Specific comments to improve the manuscript are given below.
Abstract
Lines 15-16: The sentence that begins with “Malaria cases in these districts…” can be written more clearly. Do you mean to say that many people die without showing symptoms of malaria, or without being diagnosed with malaria? The phrase “without suspecting for” sounds a bit awkward.
Line 19: I don’t think you need the word ‘credible’ at the beginning of this sentence. Why not say “Confidence intervals from a negative binomial model….”
Introduction
The Introduction is too long and contains a lot of unnecessary information. The authors are advised to drastically cut this section to 2-3 paragraphs. Paragraphs 1, 2, 5, and 8 could be deleted or cut by 95%. Paragraph 8 on page 3 is never included in a journal article.
In line 60 (and other places), it is not correct to write the citation number as in “According to [7]…”. Please change this to “According to Blumberg and Frean (2017)…”. Please review and adhere to the journal’s requirements for formatting. Other areas that need to be change include lines 75 and 76 (page 2) and throughout the paper.
Materials and Methods
This section is also too long. It is not necessary to include all of the statistical equations in the text, although it may be necessary for a Statistics journal. Please remove all of these equations from this section and place in a separate file (e.g., an appendix or supplementary file) to be submitted along with the manuscript.
Please include more details about the variables in the study (i.e., rainfall, temperature). How were these data collected and by who?
What statistical software was used to analyze the data?
Results
Include the definitions of the abbreviations in Table 2 in footnotes (at the bottom of the table).
I would recommend placing all tables at the end of the paper, after the references.
Discussion
Describe the limitations and strengths of your study.
Round 2
Reviewer 2 Report
The authors have responded to all of my comments and suggestions in a satisfactory manner. The manuscript is much improved. One minor comment. In the Abstract, change "without suspecting for malaria" to match the new changes in the text.
